# A Personalized Neoantigen Vaccine in Combination with Platinum-Based Chemotherapy Induces a T-Cell Response Coinciding with a Complete Response in Endometrial Carcinoma

**DOI:** 10.3390/cancers13225801

**Published:** 2021-11-18

**Authors:** Alexandre Harari, Apostolos Sarivalasis, Kaat de Jonge, Anne-Christine Thierry, Florian Huber, Caroline Boudousquie, Laetitia Rossier, Angela Orcurto, Martina Imbimbo, Petra Baumgaertner, Michal Bassani-Sternberg, Lana E. Kandalaft

**Affiliations:** 1Center of Experimental Therapeutics, Department of Oncology, Lausanne University Hospital (CHUV), 1011 Lausanne, Switzerland; Kaat.De-Jonge@chuv.ch (K.d.J.); Anne-Christine.Thierry@chuv.ch (A.-C.T.); florian.huber@chuv.ch (F.H.); Caroline.Boudousquie@chuv.ch (C.B.); Laetitia.Rossier@chuv.ch (L.R.); Petra.Baumgartner@hospvd.ch (P.B.); Michal.Bassani@chuv.ch (M.B.-S.); 2Ludwig Institute for Cancer Research, University of Lausanne (UNIL), 1005 Lausanne, Switzerland; 3Department of Oncology, Lausanne University Hospital (CHUV), 1011, Lausanne, Switzerland; Apostolos.Sarivalasis@chuv.ch (A.S.); Angela.Orcurto@chuv.ch (A.O.); Martina.Imbimbo@chuv.ch (M.I.)

**Keywords:** endometrial cancer, cancer vaccines, immunotherapy, neoantigens

## Abstract

**Simple Summary:**

We investigated the feasibility and immunogenicity of an autologous Dendritic Cell (DC) vaccine pulsed with peptide neoantigens in combination with a standard of care regimen. The vaccine program took place in a serous endometrial mismatch repair (MMR) deficiency setting. We demonstrate for the first time the feasibility of producing a peptide DC vaccine in endometrial carcinoma. The safety and immunogenicity of this personalized vaccine was demonstrated by the detection of polyfunctional and durable T-cell responses.

**Abstract:**

Endometrial cancer (EC) is a common gynecological malignancy and the fourth most common malignancy in European and North American women. Amongst EC, the advanced serous, p53-mutated, and pMMR subtypes have the highest risk of relapse despite optimal standard of care therapy. At present, there is no standard of care maintenance treatment to prevent relapse among these high-risk patients. Vaccines are a form of immunotherapy that can potentially increase the immunogenicity of pMMR, serous, and p53-mutated tumors to render them responsive to check point inhibitor-based immunotherapy. We demonstrate, for the first time, the feasibility of generating a personalized dendritic cell vaccine pulsed with peptide neoantigens in a patient with pMMR, p53-mutated, and serous endometrial adenocarcinoma (SEC). The personalized vaccine was administered in combination with systemic chemotherapy to treat an inoperable metastatic recurrence. This treatment association demonstrated the safety and immunogenicity of the personalized dendritic cell vaccine. Interestingly, a complete oncological response was obtained with respect to both radiological assessment and the tumor marker CA-125.

## 1. Introduction

Endometrial cancer (EC) is the most common malignancy that affects the female reproductive system. In 2018, 121,500 new cases have been estimated in Europe [1].

One-fifth of the cases are diagnosed at an advanced stage (stage III or IV), for which the 5-year overall survival is only 68% and 17%, respectively [2]. Between 10% and 19% of endometrial carcinomas are considered high-risk tumors. These tumors can be subtyped into either high-grade endometrioid or non-endometrioid tumors. These have a high potential for lymphovascular and hematogenous metastatic spread and thus account for the majority of deaths from endometrial cancer [3]. Recent developments in the molecular classification of endometrial carcinomas by the the cancer genome atlas (TCGA), separated endometrioid carcinomas in four molecular subtypes with limited overlap: polymerase epsilon (POLE), microsatellite instable (MSI), copy number high (p53-mutated), and copy number low. Of these subtypes, the copy number high, p53-mutated, accounts for 20–25% of all EC. This subgroup of p53-mutated EC is often associated with similar prognosis and common clinical features as serous endometrial carcinoma (SEC). When compared to other molecular EC subtypes, this subgroup exhibits the worst prognosis both in terms of progression-free overall survival and overall survival [4].

Currently, there is no tailored adjuvant nor metastatic treatment for advanced serous, p53-mutated, endometrial adenocarcinomas. The standard of care adjuvant therapy for high risk and advanced endometrial cancers consists of chemotherapy, radiotherapy and brachytherapy. Despite optimal adjuvant treatment, SEC and p53-mutated tumors present a high risk of relapse. At present, there is no standard of care maintenance treatment to prevent relapse among these high-risk patients. At relapse and for the treatment of metastatic p53-mutated EC or SEC, chemotherapy-based strategies are the standard of care (SoC). Notably without any maintenance treatment options.

Recent advances in immunotherapy, especially by checkpoint inhibitors (CPI), demonstrated promising results for the treatment of gynecological malignancies, especially for deficient mismatch repair (dMMR)/MSI-H tumors [5]. A second phase II trial [6] has reported promising activity of CPI based immunotherapy in combination with lenvatinib, a multi-targeted, anti-angiogenic tyrosine kinase inhibitor (TKI). This association although more effective for the treatment of dMMR tumors reported activity also in mismatchrepair proficient (pMMR) and interestingly SEC. This combination, although promising, is associated with relevant toxicities and has yet to be tested as a frontline treatment (keynote 775) [6]. Thus, the field of SEC and, more broadly, p53-mutated EC is in dire need for novel therapeutic approaches.

Tumor-infiltrating lymphocytes (TILs) have been detected in cervical and endometrial tumors with prognostic significance, suggesting that endometrial cancer is a suitable candidate for immunotherapy [7,8]. Several immunotherapeutic approaches have been tried in endometrial cancer, such as PD-L1-targeting antibodies, Bispecific T-cell engager (BiTE) antibodies, adoptive cell transfer (ACT), and cancer vaccines [9]. Unfortunately, most early therapeutic cancer vaccines studies have shown limited efficacy in endometrial cancer [10,11,12,13] and gynecological malignancy tumors [14].

One of the basic requirements to improve vaccine efficacy is the right selection of target antigens. Indeed, the reduced efficacy of the previously studied vaccines was associated with the selection of non-mutated self-antigens as targets [15,16,17,18,19,20,21]. Due to their intrinsic characteristics, such as genetic instability, tumors can generate a large number of tumor-specific mutations that lead to the expression of another class of tumor-associated antigens that could be recognized by T-cells: the mutated “neoantigens”. Deep sequencing analysis of tumor cells has revealed that they usually harbor between 10 and 1000 so-called “private somatic mutations”. The majority of these mutations induced neoantigens different even among tumors of the same histological type [22,23]. In contrast to self-antigens, T-cell reactivity towards these neoantigens demonstrated a functional avidity similar to the avidity observed in anti-viral T-cells [15]. Furthermore, T-cell responses to neoantigens is not expected to induce any autoimmune toxicity against healthy tissues, making vaccination toward neoantigens a very attractive option.

Dendritic cells are professional antigen-presenting cells capable of presenting antigen from pathogen or foreign origin in order to elicit a potent antigen-specific immune response [24]. The antigen source is a very important aspect determining the efficacy of DC-based vaccines. Targeting tumor-specific neoantigens has the advantage of attacking only the tumor and sparing the healthy tissue [25]. A recent pilot clinical trial was carried out at the Washington University School of Medicine as a “proof-of-concept” for mutanome DC vaccines. They first demonstrated that vaccination with DCs pulsed with neoantigen peptides is safe and effective in boosting neoantigen-specific T-cells in three melanoma patients [26]. Multiple phase I studies were conducted at the Dana-Farber Cancer Institute in Boston [27,28] and at the University of Mainz [29]. Together, they showed that neoantigen targeting vaccines were feasible and well tolerated without any serious side effects. In melanoma patients, neoantigen reactivity could be observed in all vaccinated patients. For all these patients, more than 30% of neoantigen-specific CD4+ and CD8+ T-cells were polyfunctional [27]. A second study including melanoma patients showed development of neoantigen-specific T-cell responses in all patients, and vaccine-induced T-cell infiltration and neoantigen-specific killing of autologous tumor cells in post-vaccination resected metastases from two patients [29]. Our group has a longstanding expertise in vaccines using autologous tumor lysates [30] in both human and mouse models. No animal model was established for the current peptide vaccine given the well-established safety of both DC vaccines and peptide vaccines.

Accumulating evidence from recent studies associates clinical benefit from immunotherapy with specific responses to private tumor epitopes [31,32,33,34,35]. However, currently, the feasibility of a private antigen-based vaccine in tumors with intermediate–low mutational burden (less than 100 mutations per tumor) is unclear. Locally advanced, stage III to IVA and recurrent metastatic, endometrial cancer exhibits such characteristics. These clinical situations are at high risk for local and metastatic recurrence and progression, despite treatment according to current standard of care including surgery, chemotherapy, and eventually, radiotherapy. Metastatic disease is often incurable, and palliative treatment is the only option available for the majority of these patients. Therefore, in this high-risk setting, a novel, personalized treatment approach is urgently needed. No neoantigen vaccine studies have been conducted in endometrial cancer patients.

We describe here for the first time the feasibility of producing and administrating of a personalized vaccine created from autologous monocyte-derived DCs (moDCs) pulsed with personalized peptides for a patient with a metastatic recurrent pMMR, SEC. We also demonstrate its safety and immunogenicity. The personalized vaccine was administered in combination with systemic chemotherapy to treat an inoperable metastatic recurrence led to complete oncological response with respect to both radiological assessment and the tumor marker CA-125.

## 2. Materials and Methods

### 2.1. Patient Clinical Protocol

The work described was conducted in accordance with the Declaration of Helsinki and approved by the ethics committee of the canton of Vaud, Switzerland and the Swiss health authority. The design of the dendritic vaccine program is summarized in Figure 1. Before enrollment in this dendritic vaccine program, the patients’ SoC treatment consisted of primary cytoreductive surgery and sequential chemotherapy. Collection of surgically debulked endometrial tumor tissue as well as blood collection and identification personalized peptides (PEP) (including quality control (QC) release) are performed under a research protocol (2016-02094) with a separate informed consent form.

Once the neoantigen prediction was successful, enrolment was proposed to the patient. Written consent for this compassionate program (TA_2019-00004) was obtained. Once the patient is registered, she underwent a leukapheresis and production of the personalized vaccine commenced. The PEP-DC vaccines were manufactured and formulated under GMP-compliant regulations in the Cell Manufacturing Facility of the Center of Experimental Therapeutics (CTE, Lausanne) at the Centre Hospitalier Universitaire Vaudois (CHUV, Lausanne).

### 2.2. Vaccine Manufacturing

Autologous monocytes were enriched in a CliniMACS Prodigy (Miltenyi Biotec, Bergisch Gladbach, North Rhine-Westphalia, Germany) system, that is, a GMP-compliant closed system, allowing for standardized and reproducible cellular processing across multiple instruments. The leukapheresis bag was attached using sterile tubing set to the CliniMACS Prodigy device, and with the predefined LP-14 enrichment program and the CD14 reagent (magnetic beads, Miltenyi Biotec), CD14+ monocytes were purified by positive enrichment. Purified monocytes were differentiated into immature monocyte-derived DC (iDC) by 5-day culture in the presence of clinical grade IL-4 (CellGenix, Freiburg im Breisgau, Baden-Württemberg, Germany) and GM-CSF (CellGenix). iDC were then loaded with a mix of the 9 long peptides overnight and matured/activated for 6–8 h on day 6 using a maturation cocktail consisting of clinical grade monophosphoryl lipid A (MPLA, Avanti, Birmingham, Alambama, United States) and IFNγ (Imunkin, Boehringer Ingelheim, Ingelheim, Rhineland-Palatinate, Germany). Finally, cells were harvested and cryopreserved as vaccine doses, comprising 5–10 × 10^6^ cells per dose. For each injection of the PEP-DC vaccine, one dose was thawed, washed, and resuspended in NaCl (B. Braun Medical, Melsungen, Hesse, Germany) 0.9% supplemented with 1% human albumin (Octapharma, Lachen, Schwyz, Switzerland) before being transferred into syringes and stored at 2–8 °C until administration to patients as subcutaneous injection. Administration, every 4 weeks, beyond progression in combination with second line, platinum-based chemotherapy was allowed. Extra vaccines available could be administered alone every 4 weeks after chemotherapy treatment completion. Safety follow-up was 30 days until end of treatment visit.

### 2.3. Peptide Neoantigen Identification

Based on the patient’s own tumor, neoantigens for the PEP-DC vaccine were determined, identified, and produced from formalin-fixed, paraffin-embedded (FFPE) tumor specimen and blood samples. The blood samples are used for peripheral blood mononuclear cells (PBMC) isolation. PBMC-derived DNA was extracted for whole genome sequencing (WES) and for molecular HLA-typing. DNA was extracted from the FFPE tumor tissue for WES. WES was performed as previously described [36]. The WES data was analyzed using our NeoDisc pipeline as previously described, and genomic variants affecting coding genes that were present in the tumor and absent from the corresponding blood were assumed somatic. Neoantigens predicted to bind to HLA-I and HLA-II molecules of the patient were prioritized and optimally long sequences were designed to cover these predicted neoantigens. Eventually, 9 target sequences with ‘enhanced quality’ peptides were manufactured at Almac (Edinburgh, UK) per Swissmedic regulations and manufacturer’s instructions and were used to formulate the PEP-DC vaccine.

### 2.4. Peptide Research Grade Synthesis

Peptides were produced by the Peptides and Tetramers Core Facility (PTCF) of the University Hospital of Lausanne (CHUV). Each peptide was HPLC purified (>90% purity), verified by mass spectrometry, and kept lyophilized at −80 °C. Before usage, the peptides were resuspended in DMSO at a concentration of 10mM for long-term storage at −20 °C. For ex vivo experiments, peptides were diluted 1:10 in DMSO and used at a final concentration of 1µM.

### 2.5. Peripheral Blood Mononuclear Cells Isolation and Cryopreservation

PBMCs were isolated from blood by density gradient using Ficoll-Paque-Plus (GE Healthcare, Chicago, IL, USA) according to the laboratory standard SOP and immediately cryopreserved in 90% FCS (Fetal Calf Serum; Gibco, Thermo Fisher Scientific, Waltham, MA, USA) and 10% DMSO (dimethyl sulfoxide, SIGMA, Merck KGaA, Darmstadt, Hesse, Germany). Cryo-preserved PBMC were thawed in RPMI (Invitrogen, Thermo Fisher Scientific, Waltham, MA, USA) 20% FCS.

### 2.6. Flow Cytometry Analyses

After thawing, PBMC were washed and immediately stained with the following panel: anti-CD3 (clone UCHT1; #IM2467, Beckman Coulter, Brea, CA, USA), anti-CD4 (clone SFCI12T4D11; #737660, Beckman Coulter), anti-CD8 (clone RPA-T8; #558207, BD Biosciences, Franklin Lakes, NJ, USA), anti-CD14 (cloneMφP9; #641394, BD Biosciences), anti-CD16 (clone3G8; #555406, BD Biosciences), anti-CD56 (clone N901; #A07788, Beckman Coulter), anti-CD11c (clone B-ly6; #561352, BD Biosciences), anti-CD19 (clone SJ25C1; #563036, BD Biosciences), anti-CD123 (clone 6H6; #45-1239-42, eBioscience, Affymetrix, Santa Clara, CA, USA), anti-HLA-DR (clone Immu-357; #IM3636, Beckman Coulter), and Zombie UV (#77474, Biolegend, San Diego, CA, USA). The samples were acquired on an BD Fortessa instrument equipped with the FACS Diva software. The analysis was performed with the FlowJo (FLOWJO, LLC, Ashland, OR, USA).

### 2.7. In Vitro Stimulation (IVS) of Antigen-Specific T-cells

Cryopreserved PBMC were stimulated with peptides (individual or peptide pools) at a final concentration of 1µg/mL per peptide in RPMI supplemented with 8% human serum (Biowest, Nuaillé, Maine-et-Loire, France), Penicillin/Streptomycin (BioConcept, Allschwil, Basel-Country, Switzerland) and beta-mercaptoethanol (Gibco). Recombinant human IL-2 (Proleukin, Clinigen, Yardley, Pennsylvania, United States) was added to the culture after 48 h at a final concentration of 100 U/mL for 12 days. At day 12 of the IVS, IFNγ Enzyme-Linked ImmunoSpot (ELISpot) assays were performed using pre-coated 96-well ELISpot plates (3420-2APT-10 Mabtech, Stockholm, Södermanland and Uppland, Sweden) [37]. Then, 100,000 cells were re-challenged (in duplicate) with peptides for 16–18 h, and then plates were washed according to manufacturer’s instructions and counted with the AID-Spot Robot ELISpot reader (AutoImmun Diagnostika GMBH, Straßberg, Baden-Württemberg, Germany). SEB (Staphylococcal Enterotoxin B, final concentration of 250ng/mL, Sigma-S4881) and anti-CD3 antibody (final concentration of 1 µg/mL, clone UCHT1, Mabtech) were used as positive control and unstimulated PBMCs as background control.

### 2.8. Peptide Dose Response Curve by IFNγ ELISpot Assay

At day 12 of the IVS, IFNγ ELISpot assays were performed using pre-coated 96-well ELISpot plates (Mabtech #3420-2APT-10) [37]. Then, 100,000 cells were re-challenged either with the mutated or wild type sequence of cognate peptides used at different concentrations (2 µM, 1 µM, 0.1 µM, and 0.001 µM) for 16–18 h. Plates were then washed according to manufacturer’s instructions and counted with the AID-Spot Robot ELISpot reader (AutoImmun Diagnostika GMBH).

### 2.9. Flow Cytometry Analysis after In Vitro Stimulation

To determine by flow cytometry the phenotype and cytokine profiles of vaccine-specific CD4+ and CD8+ T-cells 12 days after in vitro stimulation (IVS), patient PBMCs were re-challenged with individual peptides overnight at 37 °C, 5%CO2 in presence of Brefeldin A (51-2301KZ, BD Bioscience). The next day, the cells were washed with PBS and stained for 20 min on ice with anti-CD3 (clone SK7; # 344840, Biolegend), anti-CD8 (clone RPA-T8; #301042, Biolegend), anti-CD4 (clone RPA-T4; #558116, BD Biosciences), and a viability dye Zombie UV (#77474, Biolegend). After fixation and permeabilization for 20 min at 4 °C (BD Bioscience Cytofix/Cytoperm Kit), anti-CD154 (CD40-L, clone 24–31; #310826, Biolegend), anti-Granzyme B (clone GB11; #GRB17, Invitrogen), anti-Perforin (clone B-D48; #353310, Biolegend), anti-IL-2 (clone MQ1-17H12; #559334, BD Biosciences), anti-TNF-α (clone MAb11; #557647, BD Biosciences), and anti-IFN-γ (clone B27; #554702, BD Biosciences). All samples were acquired on a 5-laser BD FORTESSA instrument equipped with the FACS DiVa software. Analyses were performed with FlowJo v10.5 (FLOWJ,.LLC, Ashland, OR, USA) and SPICE 6 software.

## 3. Results

### 3.1. Clinical Efficacy

The first patient included in our dendritic vaccination program was 43 years suffering a relapsed high-grade, p53 mutated, serous endometrial carcinoma. The disease stage was FIGO IIIC2, pT3 pN2a (16/31) cM0 with extensive lymphovascular space invasion. The CA-125 value was 1198kU/L (UNL < 35 kU/L). The disease was pMMR, HER2 negative by immunohistochemistry, expressed PD-L1 at 1% and showed evidence of homologous recombination deficiency (HRD) on the Oncoscan while germline BRCA1/2 mutational status was negative.

The patient underwent a total radical hysterectomy, with bilateral oophorectomy, omentectomy, peritoneal cytology, and pelvic and para-aortic lymph node dissection by median laparotomy, followed by a sequential chemotherapy and radiotherapy treatment by six cycles of carboplatinum AUC5 and paclitaxel 175 mg/m^2^ adjuvant treatment followed by radiotherapy. The patient was then in complete remission, with CA-125 value of 16 kU/L, and was enrolled in the vaccination program to receive a personalized dendritic cell vaccine pulsed with peptide neoantigens (PEP-DC) in combination with standard of care therapy at the discretion of the investigator (Figure 1). The patient received three cycles of PEP-DC. The patient experienced a radiological and biochemical relapse, localized to the previous irradiated left internal iliac lymph nodes and the mesorectal lymph nodes, 11 months since her primary treatment. Her CA-125 rose to 158kU/L. Carboplatinum AUC5 in combination with liposomal pegylated doxorubicine (PLD) at 30mg/m^2^ was added to her vaccination schedule every 4 weeks. After five cycles of chemotherapy and five PEP-DC vaccine administrations, the patient experienced a complete remission, as demonstrated by her CT-scan and her CA-125 level (Figure 2A–C). Thirteen months since her last dose of chemotherapy, the patient is still (as of September 2021) in complete radiological and laboratory response. Two additional vaccines were administered after the completion of the chemotherapy treatment. Two grade three hematological adverse events were observed; unrelated to the vaccine, but related to the chemotherapy. The observed hematological grade three toxicities concerned neutropenia occurring on day 8 and day 7 of the fourth and fifth chemotherapy cycles. Overall, the association was well tolerated without unexpected toxicities, notably immune related ones.

### 3.2. Longitudinal Analysis of Main Circulating Immune Cell Subsets

Peripheral blood was collected at eleven time points during the clinical vaccination program period. Cryopreserved PBMCs were then thawed in batches and labelled with an antibody panel to capture main circulating immune cell subsets [38]. Only minor changes in frequency were observed throughout the vaccination period for most T-cell populations (total T and B cells, CD4 and CD8 T-cells, NK subsets, DC subsets, and monocytes subsets, Figure 3A).

### 3.3. Vaccine Immunogenicity

We next assessed vaccine immunogenicity. To this end, PBMCs collected before vaccination and after three vaccinations (vaccine 3, Vax3) were interrogated for the presence of T-cells recognizing the vaccine (i.e., the pool of nine peptides composing the vaccine) (Table 1). Of interest, a high frequency of T-cells recognizing the vaccine (pool of peptides) was detected in post vaccination but not in pre-vaccination T-cells (Figure 3B), indicating the lack of pre-existing cellular immune response against any of the peptides from the vaccine as well as a potent vaccine antigenicity. Further, we interrogated post-vaccination PBMCs with every single long peptide present in the vaccine and identified two responses against the long peptides covering mutations in TGIF2 and ALDH1B1 (Figure 3C and Table 1). T-cell responses against TGIF2 and ALDH1B1 were assessed at the end of the vaccination program and were potent (>3000 SFU/10^6^ PBMC for both epitopes) (Figure 3C).

### 3.4. Mapping of Tumor-Specific T-Cell Responses

The proteogenomic analyses on tumor and germline material leads to an extensive list of potential antigens, ranked according to multiparametric criteria. Top predicted antigens were selected for vaccination. In addition, we also determined a panel of 73 additional private tumor neoantigens that we selected to interrogate pre- and post-vaccination immunity to assess see antigen spreading (absent in the vaccine, Table 2). Of interest, we only identified one clear neoepitope among the 73 candidates (MYH14, Table 2) (Figure 3D). Of note, T-cell responses against each of the three neoepitopes (TGIF2, ALDH1B1, and MYH14) were specific, as no, or very limited, cross-reactivity was detected against any of the cognate wild-type peptides (Figure 3E). Furthermore, intracellular cytokine staining experiments by flow cytometry indicated that T-cell responses against TGIF2, ALDH1B1 and MYH14 were all mediated by CD4 T-cells (Figure 3F).

### 3.5. Kinetic of Tumor-Specific CD4 T-Cell Responses

Of interest, whereas MYH14-specific CD4 T-cells were already detectable at prevaccination, their frequency significantly and transiently increased shortly after vaccinations (Figure 3G). The frequencies of TGIF2 and ALDH1B1-specific CD4 T-cells also peaked after three vaccinations and then remained stable until the end of study (EOS). Of note, the magnitude of MYH14-specific T-cells at EOS (<1000 SFU/106 PBMC) was lower than that of each of the two vaccine peptides (>3000 SFU/106 PBMC for each vaccine peptide) (Figure 3C,D) and was confirmed in flow cytometry experiments (Figure 3G).

### 3.6. Functional Profile of Vaccine-Specific CD4 T-Cell Responses

Having established the magnitude, specificity, and kinetics of vaccine-specific CD4 T-cell responses, we finally investigated whether repeated vaccinations impacted the functional profile of vaccine-specific T-cells. Using polychromatic flow cytometry, we assessed the functional profile of vaccine-specific CD4 T-cell responses by measuring the proportion of TGIF2-specific CD4 T-cells able to mediate one or multiple functions (such as cytokines production or expression of effector molecules). Of interest, TGIF2-specific CD4 T-cells from Vax6 were significantly more polyfunctional than those collected at Vax2, despite an overall similar frequency of circulating cells between the two timepoints (Figure 3H).

Taken together, these observations indicate that the vaccine is immunogenic and can induce potent, polyfunctional, and durable T-cell responses, which coincide with a complete radiological response and a normalization of the tumor marker CA-125.

## 4. Discussion

The clinical translation of cancer vaccines into efficacious therapies is challenging. One of the reasons for low therapeutic efficacy of many previous vaccines is their inability to elicit a rapid and strong T-cell response [21]. Furthermore, even when a defined target is known, and the vaccination induces an immune response, evolution of antigen expression and the loss of its expression by the process of immune editing may diminish long-term benefits [39]. This suggests that targeting multiple antigens as vaccine targets and combination therapies targeting multiple pathways along with diminishing the immunosuppressive microenvironment may prove synergistic and may achieve significant clinical benefits.

We describe here for the first time the feasibility of producing a personalized vaccine created from autologous moDCs pulsed with personalized neoantigenic peptides. The first patient included in our dendritic vaccine program was 43 years old, suffering a relapsed high-grade, p53 mutated, serous endometrial carcinoma. The concomitant treatment of platinum-based chemotherapy with our PEP-DC vaccination was shown to be feasible and safe. The main observed side effects concerned grade 3 hematological toxicity. After five cycles of chemotherapy and nine vaccine doses, the patient had a complete response with respect to both radiological assessment and CA-125 tumor marker. The patient continued on a maintenance vaccination program and received two additional vaccine doses. The patient remained in complete remission a year after the last vaccine dose. The addition of a concurrent personalized neoantigen vaccine to the platinum-based chemotherapy lead to a significant clinical benefit for our patient. This second-line treatment association resulted in a long lasting complete biochemical and radiological response. Notably, the progression-free survival (PFS) 2, of 13 months, outlasted the PFS1 of 11 months of the primary treatment. This remission inversion is clinically significant and relevant especially in this challenging oncological setting of a recurrent, non-debulkable, SEC. Furthermore, when compared with historical PFS data of chemotherapy-exclusive treatment, where PFS vary according to treatment sequence between 3 and 10 months [40], the results of the treatment association of our patient resulted in long lasting complete response of 13 months with a response that is still ongoing.

Moreover, the vaccine was shown to be immunogenic, as T-cells isolated from the PBMCs from the patient after vaccination were able to recognize the vaccine peptides. More specifically, two neoepitopes covering mutations in TGIF2 and ALDH1B1 genes were identified. T-cell responses directed against one additional neoepitope, MYH14, not present in the vaccine increased after vaccination. T-cell responses against these targets were mediated by CD4 T-cells. Moreover, with an increasing number of vaccinations, T-cells became more polyfunctional. Taken together, these observations indicate that the vaccine is immunogenic and can induce potent, polyfunctional, and durable T-cell responses, which also coincide with clinical benefit in a very difficult disease setting.

## 5. Conclusions

To our knowledge, this study is the first to report on feasibility, safety, and efficacy of a concomitant treatment associating a platinum-based chemotherapy with a PEP-DC vaccine in context of a recurrent SEC. As reported, the treatment combination was feasible and safe without unexpected or added toxicity in comparison to the known toxicities expected form the carboplatinum–liposomal pegylated doxorubicine. Furthermore, this association between chemotherapy and therapeutic vaccine elicited immune response as demonstrated by polyfunctional and durable T-cell responses. Moreover, this combination lead to a long lasting oncological complete response in this otherwise challenging recurrent metastatic setting. Of note, these results outperformed the 1st line treatment PFS. Thus, this treatment association is promising and should be further investigated in the current and other potential settings. 

## Figures and Tables

**Figure 1 cancers-13-05801-f001:**
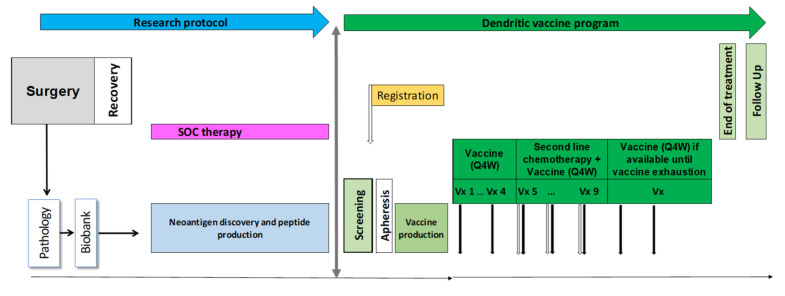
Design of the dendritic vaccine program.

**Figure 2 cancers-13-05801-f002:**
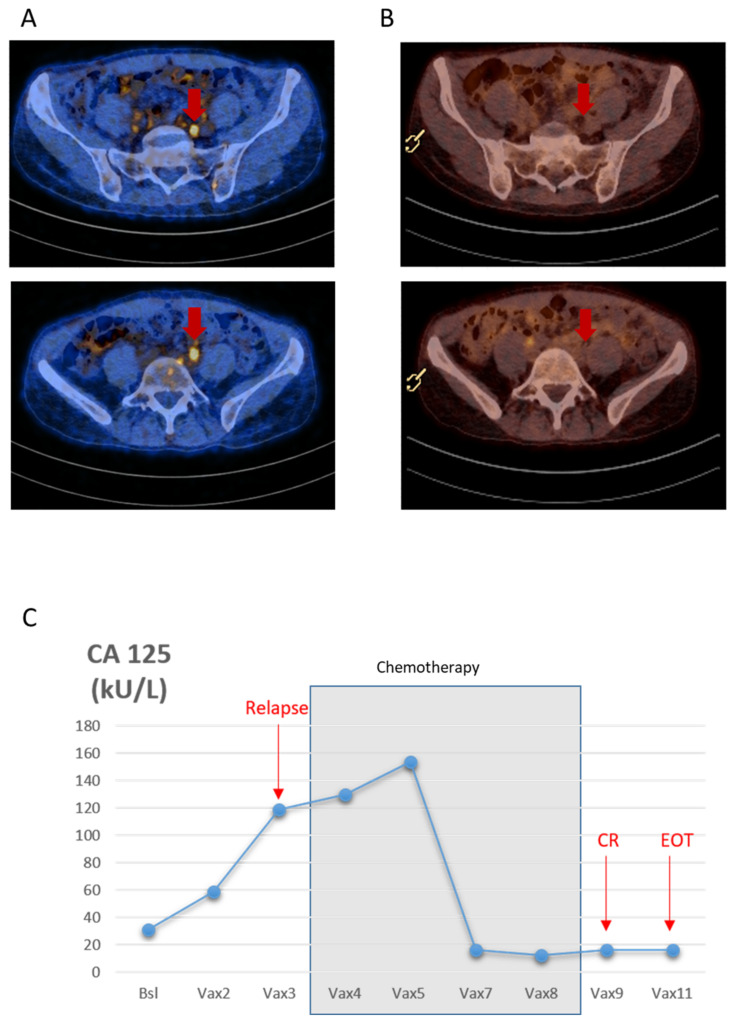
(**A**) PET-CT images showing three lymph nodes with pathological tracer uptake with a small diameter (11 mm, 9 mm and 6 mm). (**B**) PET-CT images showing complete radiological response. (**C**) Longitudinal representation of CA-125 values from baseline (November 2019) to relapse (February 2020) and complete response (CR) (July 2020) until end of treatment (EOT) (October 2020) The chemotherapy treatment period is indicated in grey.

**Figure 3 cancers-13-05801-f003:**
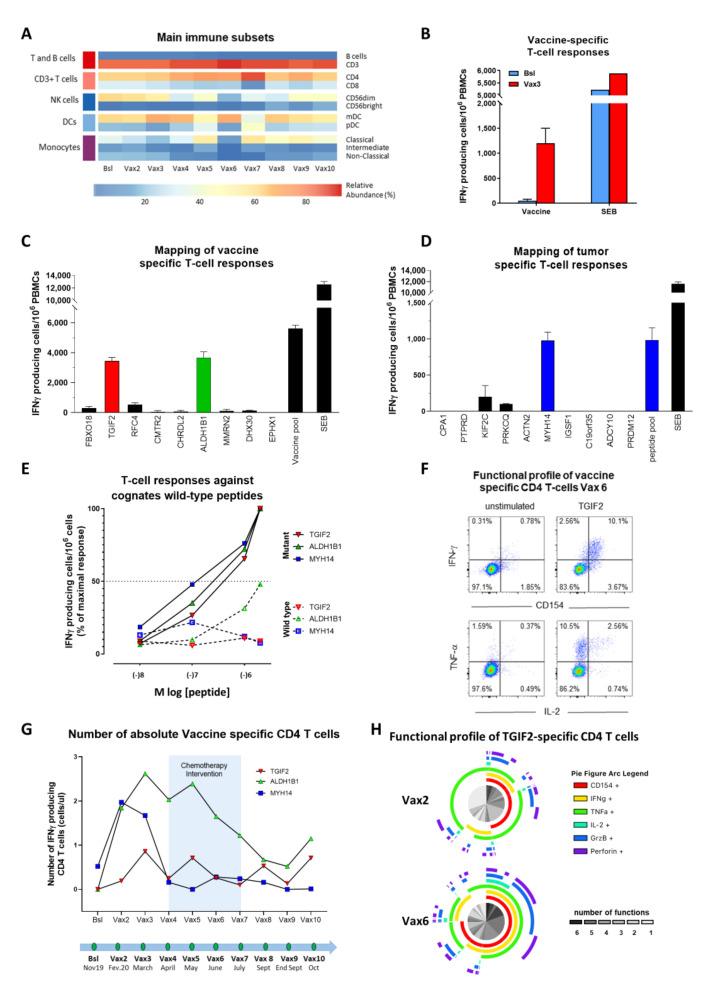
Immune profiling of vaccine T-cell response. (**A**) Heatmap representation of relative frequencies of ten main cell subsets in peripheral blood over the DC vaccine program period. (**B**) Vaccine-specific T-cell responses (IFNγ-ELISpot) of PBMCs at baseline (BSL) and after three vaccinations (post-Vax3) post-IVS performed with the vaccination peptide pool (Vaccine). Background response of unstimulated cells were subtracted and SEB was used as positive control. (**C**) Deconvolution (IFNγ-ELISpot) of the nine individual peptides included in the vaccination peptide pool (vaccine) post-IVS. Post Vax6 PBMC were used. Background signal of unstimulated cells were subtracted and SEB was used as positive control. (**D**) Deconvolution (IFNγ-ELISpot) of ten predicted peptides of the positive responding peptide sub-pool after IVS on PBMCs from time point post Vax6. Background response of unstimulated cells were subtracted and SEB was used as positive control. (**E**) Dose response curve (IFNγ-ELISpot) of identified responding mutated vaccination peptides TGIF2, ALDH1B1 and predicted peptide MYH14 compared to their corresponding wild type peptides. Post Vax6 PBMC were used. (**F**) Representative example of cytokine production of CD4 T-cells after re-challenge with TGIF2 peptide at post Vax6. (**G**) Longitudinal analysis of the number of peptide-specific IFNγ-secreting CD4 T-cells per μl of blood. The light blue area represents the period of chemotherapy from post Vax4 to post Vax7. (**H**) Functional profile of TGIF2-specific CD4 T-cells collected at Vax2 and Vax 6. (**B**–**D**) Error bars represent the standard deviations.

**Table 1 cancers-13-05801-t001:** List of neoantigen peptides included in the vaccine.

Gene	Mutation	Sequence
*FBXO18*	p.Asn183His	APPSRKRSWSSEEESHQATGTSRWD
** *TGIF2* **	**p.Glu203Gln**	**KEDFSSFQLLVQVALQRAAEMELQK**
*RFC4*	p.Ala279Pro	DIAGVIPAEKIDGVFPACQSGSFDK
*CMTR2*	p.Glu446Lys	KWFGQRNKYFKTYNKRKMLEALSWK
*CHRDL2*	p.His403Leu	DFQKEAQHFRLLAGPLEGHWNVFLA
** *ALDH1B1* **	**p.Ala153Val**	**LDEVIKVYRYFAGWVDKWHGKTIPM**
*MMRN2*	p.His319Gln	DVEDRLHAQQFTLHRSISELQADVD
*DHX30*	p.Ile873Met	SKAVDSPNMKAVDEAVILLQEIGV
*EPHX1*	p.Leu172Arg	KNHGRSDEHVFEVICPSIPG

**Table 2 cancers-13-05801-t002:** List of additional neoantigen peptides.

**Gene**	**Mutation**	**Sequence**	**Pool Number**
*BCAN*	p.Gly648Val	VPASVNSAQGSTALSILLL	**1**
*DIAPH1*	p.Cys267Tyr	VPNMMIDAAKLLSALYIL	
*CNTNAP2*	p.Arg389Trp	VPVFFNATSYLEVPGWL	
*NDN*	p.Asp321Gly	EANPTAHYPRSSVSEG	
*LPCAT1*	p.Cys514Phe	GFFADFSPENSDAGRK	
*TXNRD2*	p.Glu413Gln	VGLSEQEAVARHGQEH	
*SLC2A4*	p.Ser281Arg	LGRRTHRQPLIIAVVL	
*ALS2*	p.Pro1029Arg	LPPYGSGSSVQRQEPR	
*ATAD2B*	p.His1097Gln	RGLSVTSEQINPQSTG	
*SYT7*	p.Asp642Asn	LRETTIIITVMDKNKL	
*CPA1*	p.Leu96Pro	HGISYETMIEDVQSPL	**2**
*PTPRD*	p.Asp1248Glu	YSEPVVSMDLDPQPIT	
*KIF2C*	p.Met217Ile	EMRIKRAQEYDSSFPN	
*PRKCQ*	p.Ile73Thr	STFDAHINKGRVMQIT	
*ACTN2*	p.Thr609Ile	SSNPYSTVTMDELRIK	
** *MYH14* **	**p.Arg861His**	**AARGYLAHRAFQKRQQ**	
*IGSF1*	p.Ser866Gly	NYSCRYYDFSIWSEPG	
*C19orf35*	p.Ala210Ser	DAWHILVAKVPKPGSD	
*ADCY10*	p.Cys953Tyr	KAMHLKYARFLEEDAH	
*PRDM12*	p.Gly21Arg	APRLALAEVITSDILH	
*TFAP2A*	p.Pro317Arg	GTHGRTPSWSPASRAA	**3**
*EPHX1*	p.Leu172Arg	RSDEHVFEVICPSIPG	
*PNPLA6*	p.Pro707Ser	SRATTVHAVRDTELAK	
*NSD1*	p.Ala976Thr	STNPSPSGGDSALSG	
*WFDC3*	p.Pro144Ser	EELCDGDASCSQGHK	
*TRAV7*	p.Gln59His	TYSVSRFNNLQWYRH	
*UBB*	p.Asp210Gly	SGYNIQKESTLHLVL	
*UBB*	p.Asp210Gly	KQLEDGRTLSGYNIQ	
*EPHX1*	p.Leu172Arg	YKIIPLLTDPKNHGR	
*EPHX1*	p.Leu172Arg	KIIPLLTDPKNHGRS	
*CHRDL2*	p.His403Leu	LEGHWNVFLAQTLEL	**4**
*DIAPH1*	p.Cys267Tyr	LSALYILPQPEDMNE	
*LPCAT1*	p.Cys514Phe	IPNGFFADFSPENSD	
*KLHL9*	p.Ala410Gly	VGGRSAGGELATVE	
*DOCK5*	p.Pro1851Ser	LPVRREAKAPSPPP	
*TRPM6*	p.Val613Leu	TGFLYPYNDLLLWA	
*KAT5*	p.Gly31Cys	GPPVADPCVALSPQ	
*KCNIP1*	p.Ala104Thr	SPTLLFCLVDASTY	
*PSG1*	p.Phe426Ile	VSGKWIPASLAIGI	
*KIAA1551*	p.Leu156Val	SQVITSDTYSMQMQ	
*USP4*	p.Asp249Glu	GMALQNYENKLVK	**5**
*CD244*	p.Tyr369Phe	SRKELENFDVF	
*UBB*	p.Asp210Gly	QLEDGRTLSGY	
*AS3MT*	p.Ile274Thr	VTYNGGITGH	
*CSMD3*	p.Asp3578Tyr	KMKEENWAMY	
*MYH14*	p.Arg861His	AARGYLAHR	
*CNTNAP2*	p.Arg389Trp	SYLEVPGWL	
*UBB*	p.Asp210Gly	GRTLSGYNI	
*DIAPH1*	p.Cys267Tyr	KLLSALYIL	
*ASH1L*	p.Ser1539Ala	CHMACPHLS	
*UBR1*	p.Met1051Leu	LLYDNTSEM	**6**
*RALGAPB*	p.Arg1130His	HLPPHLIAL	
*KIAA1683*	p.Thr1164Ala	HLCRATATI	
*RNF207*	p.Leu485Val	ASVEGMRVV	
*PPAT*	p.Gln106His	CHPFVVETL	
*VAV1*	p.Asn316Lys	RAKNGRFTL	
*SLC24A2*	p.Ala390Thr	KASILHKIT	
*TMC3*	p.His852Asn	THIEDVNSE	
*BIRC7*	p.Asp96His	LASFYHWPL	
*SLC5A5*	p.Asp322Glu	EQYMPLLVL	
*PSG2*	p.Leu236His	HHGPDLPRI	**7**
*LRP2*	p.Asn1280His	FHCDHGNCI	
*RPTN*	p.Gln261Arg	QASHFNRTN	
*IPO4*	p.Pro825Ser	QSLTWLHRL	
*DCST1*	p.Ser35Asn	RQKNGLLSY	
*C17orf64*	p.Tyr80Asp	KLKDMKQSL	
*UBB*	p.Asp210Gly	TLSGYNIQK	
*EPHX1*	p.Leu172Arg	HGRSDEHVF	
*DHX30*	p.Ile873Met	SKAVDSPNM	
*DHX30*	p.Ile873Met	KAVDSPNMK	
*DHX30*	p.Ile873Met	VDSPNMKAV	**8**
*DIAPH1*	p.Cys267Tyr	AAKLLSALY	
*LPCAT1*	p.Cys514Phe	IPNGFFADF	

## Data Availability

The data presented in this study are available on request from the corresponding author.

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
