# Peer review of "A Personalized Neoantigen Vaccine in Combination with Platinum-Based Chemotherapy Induces a T-Cell Response Coinciding with a Complete Response in Endometrial Carcinoma"

_cancers, 2021, doi:10.3390/cancers13225801_

Round 1

Reviewer 1 Report

Review of “A Personalised neoantigen vaccine in combination with platinum based chemotherapy induces a T cell response coinciding with a complete response in endometrial carcinoma”

Throughout the paper

  • Define all abbreviations with first use.

Simple summary

  • Avoid repeating information for conciseness. (Lines 14-18)
  • Include punctuation or divide information into separate sentences for clarity.

Introduction

  • Avoid using the same opening sentence as in the abstract. (Line 36-37)
  • Is it necessary to add ‘estimated’ new cases? (Line 37).
  • This sentence is a little long, include punctuation or divide into separate sentences. (Lines 89-93)
  • It is unclear in what you are trying to say in this sentence. (Lines 98-99)
  • This sentence seems out of place, try including it earlier in the paragraph/introduction.

Materials and Methods

  • It is unclear where vaccine manufacturing took place. Did you manufacture vaccines in a sterile environment? Eg. In a hood?
  • How was the final dosage determined? (Line 151)
  • Did you use a model organism to test these vaccines before use in this study? If so, please detail.
  • List sources of all reagents and materials, as there are quite a few missing.
  • There was no statistical analysis of data in your study, consider adding this to support your findings.

Results

  • How did you determine that the adverse hematological events were unrelated to the study? (Line 259)
  • How did you come to this conclusion without statistical analysis? Instead of estimations, a statistical test such as a paired analysis, should be conducted for measuring cell populations from one individual, over time. (Line 265-268)
  • Expand your explanation of the ‘panel of 73 additional private tumour neoantigens’.

Figures

  • Standard error bars are shown in some figures, but it is not clear what they represent. (eg. Is it standard deviation from the mean...?)
  • Figure 3.7 – The formatting could be improved for ease of interpretation by the reader. The size also needs to be substantially bigger.
  • Figure 3.B – what do the error bars represent?
  • Include statistical test to confirm significance of results. Include these in your figures and figure legends.

Conclusion

  • This information would have been more beneficial in the beginning of the paper (intro). (Lines 383-389)
  • Discuss prospects for research, significance and clinical relevance.

Author Response

Throughout the paper

  • Define all abbreviations with first use.

We thank the reviewer for this comment and we have updated the manuscript accordingly.

Simple summary

  • Avoid repeating information for conciseness. (Lines 14-18) Include punctuation or divide information into separate sentences for clarity.

We thank the reviewer for this comment and we have updated the manuscript accordingly.

Introduction

  • Avoid using the same opening sentence as in the abstract. (Line 36-37)

We thank the reviewer for this comment and we have updated the manuscript accordingly.

  • Is it necessary to add ‘estimated’ new cases? (Line 37).

We thank the reviewer for this comment; we feel that an estimation of the number of cases gives us an indication of how common the malignancy is, and thus kept this sentence.

  • This sentence is a little long, include punctuation or divide into separate sentences. (Lines 89-93)

We thank the reviewer for this comment. We divided the sentence into separate sentences.

  • It is unclear in what you are trying to say in this sentence. (Lines 98-99). This sentence seems out of place, try including it earlier in the paragraph/introduction.

We thank the reviewer for this comment. We have removed the sentence.

Materials and Methods

  • It is unclear where vaccine manufacturing took place. Did you manufacture vaccines in a sterile environment? Eg. In a hood?

We thank the reviewer for this comment, we have added the following specification in paragraph 2.2. The PEP-DC vaccines were manufactured and formulated in the under GMP-compliant regulations in the Cell Manufacturing Facility of the Center of Experimental Therapeutics (CTE, Lausanne) at the Centre Hospitalier Universitaire Vaudois (CHUV, Lausanne).

  • How was the final dosage determined? (Line 151)

The dosage was determined based on the literature and on our own experience with DC vaccines published in several papers as demonstrated in the references below:

  1. Kandalaft L.E., Powell D.J. Jr., Chiang C.L., Tanyi J., Kim S., Bosch M., Montone K., Mick R., Levine B.L., Torigian D.A., June C.H., Coukos G. Autologous lysate-pulsed dendritic cell vaccination followed by adoptive transfer of vaccine-primed ex vivo co-stimulated T cells in recurrent ovarian cancer. – Oncoimmunology. 2013 Jan 1; 2(1):e22664.
  2. Kandalaft L.E., Chiang C.L., Tanyi J., Motz G., Balint K., Mick R., Coukos G. A Phase I vaccine trial using dendritic cells pulsed with autologous oxidized lysate for recurrent ovarian cancer. – J Transl Med. 2013 Jun 18; 11:149.
  3. Kandalaft L.E., Chiang C.L., Tanyi J., Hagemann A.R., Motz G.T., Svoronos N., Montone K., Mantia-Smaldone G.M., Smith L., Nisenbaum H.L., Levine B.L., Kalos M., Czerniecki B.J., Torigian D.A., Powell D.J. Jr., Mick R., Coukos G. A dendritic cell vaccine pulsed with autologous hypochlorous acid-oxidized ovarian cancer lysate primes effective broad antitumor immunity: from bench to bedside. – Clin Cancer Res. 2013 Sep 1; 19(17):4801-15.
  4. Tanyi J, Bobisse S, Ophir E, Tuyaerts S, Roberti A, Genolet R, Racle J, Baumgartner P, Stevenson B, Iseli C, Dangaj D, Czerniecki B, Semilietof A, Xenarios I, Schmidt J, Chiang C, Vanhecke D, Levine B, Monos DS, Schuster S, Svoboda J, Torigian DA, Niessenbaum H, Michielin O, June CH, Speiser D, Powel Jr D, Mick R, Zoete V, Harari A, Coukos G, Kandalaft L.E.Personalized Cancer Vaccine Effectively Mobilizes Antitumor Immunity in Ovarian Cancer – Sci Transl Med., 2018 Apr 11;10(436).
  5. Tanyi JL, Chiang CL, Chiffelle J, Thierry AC, Baumgartener P, Huber F, Goepfert C, Tarussio D, Tissot S, Torigian DA, Nisenbaum HL, Stevenson BJ, Guiren HF, Ahmed R, Huguenin-Bergenat AL, Zsiros E, Bassani-Sternberg M, Mick R, Powell DJ Jr, Coukos G, Harari A, Kandalaft LE. Personalized cancer vaccine strategy elicits polyfunctional T cells and demonstrates clinical benefits in ovarian cancer. Npg Vaccines March 2021.  4;6(1):68.  
  • Did you use a model organism to test these vaccines before use in this study? If so, please detail.

Our group has a longstanding expertise in vaccines using autologous tumor lysates in both human and mouse models. No animal model was established for the current peptide vaccine given the well-established safety of both DC vaccines and peptide vaccines. This is now added to the introduction in order to clarify this point.

  • List sources of all reagents and materials, as there are quite a few missing.

We thank the reviewer for this comment; we have added all missing details in the materials and methods section.

  • There was no statistical analysis of data in your study, consider adding this to support your findings.

We thank the reviewer for this comment. Statistical analyses were added wherever possible and relevant in the current version of the manuscript.

Results

  • How did you determine that the adverse hematological events were unrelated to the study? (Line 259)

We thank the reviewers for this query on the relation of the hematological adverse events with the study treatment. The observed hematological adverse events were closely related to the chemotherapy administration occurring within the expected period of chemotherapy related toxicities. Thus, although we cannot completely exclude this side effect to be in relation with the study treatment, the grade, timing and evolution is unexceptional for the chemotherapy treatment.

We have added the following sentence to section 3.1 in order to provide more details about the experienced toxicities: The observed hematological grade 3 toxicities concerned neutropenia occurring on day 8 and day 7 of the fourth and fifth chemotherapy cycles.

  • How did you come to this conclusion without statistical analysis? Instead of estimations, a statistical test such as a paired analysis, should be conducted for measuring cell populations from one individual, over time. (Line 265-268)

This figure 3A represents Flow cytometry data with in average of 300’000 cells (viability of ~93%) acquired per time point. The raw data for this graph are percentages. We can make some statistics for all populations, but we do not claim changes – so does it make sense? If we say “no major changes” were observed?

  • Expand your explanation of the ‘panel of 73 additional private tumour neoantigens’.

We thank the reviewer for this comment and the opportunity to improve our manuscript. A section has been added in the revised version of the manuscript, please see section 3.4.  

Figures

  • Standard error bars are shown in some figures, but it is not clear what they represent. (eg. Is it standard deviation from the mean...?)

We thank the reviewer for this question. Error bars in Fig. 3B-D represent standard deviations. A sentence has been added in the figure legend to clarify this point.

  • Figure 3.7 – The formatting could be improved for ease of interpretation by the reader. The size also needs to be substantially bigger.

We thank the reviewer for this comment. We have increased the size of the figure to a full page and hope that this improves the readability of the figure.

  • Figure 3.B – what do the error bars represent?

We thank the reviewer for this question. Error bars in Fig. 3B represent standard deviations. A sentence has been added in the figure legend to clarify this point.

  • Include statistical test to confirm significance of results. Include these in your figures and figure legends.

We thank the reviewer for this comment. Statistical analyses were added wherever possible and relevant in the current version of the manuscript.

Conclusion

  • This information would have been more beneficial in the beginning of the paper (intro). (Lines 383-389)
  • We thank the reviewer for this comment. This information has been transferred to the introduction.
  • Discuss prospects for research, significance and clinical relevance.

We thank the reviewer for their comment; we have completely rewritten the conclusion to better discuss these points.

To our knowledge, this study is the first to report on feasibility, safety and efficacy of a concomitant treatment associating a platinum-based chemotherapy with a PEP-DC vaccine in context of a recurrent SEC. As reported, the treatment combination was feasible and safe without unexpected or added toxicity in comparison to the known toxicities expected form the carboplatinum –liposomal pegylated doxorubicine. Furthermore, this association between chemotherapy and therapeutic vaccine elicited immune response as demonstrated by polyfunctional and durable T cell responses. Moreover, this combination lead to a long lasting oncological complete response in this otherwise challenging recurrent metastatic setting. Of note these results outperformed the 1st line treatment PFS. Thus, this treatment association is promising and should be further investigated in the current and other potential settings.

Reviewer 2 Report

This is an exciting paper demonstrating the use of a personalised dendritic cell vaccine used with peptide antigens in a patient with endometrial carcinoma.  The idea is novel and represents a promising new treatment paradigm for metastatic endometrial cancer.

Specific comments

  1.  It appeared that the vaccine did not prolong time to recurrence as the patient recurred soon after primary treatment despite the administration of the vaccine.  However, a complete response was seen when used with chemotherapy.  Would the authors attempt to explain this and how would this correlate with the immune response generated ?
  2. In the results presentation, it would be better if the authors could describe in more details on the sites of recurrence and the size of the recurrent lesions. The PET-CT images in Fig 2A  showed a snap shot of the sites of recurrence but it would be better if these could be described in the text.  In addition, the background colour for Fig 2A and 2B were different,  and this would make the results less convincing.  The anatomical structures that the arrows were pointing to also needed to be double checked. 
  3. More description of the toxicities experienced by the patient would be useful.  
  4. For Fig 2C, it would be helpful to indicate when the chemotherapy was given as well, so that the reader could have some better idea of how CA 125 response correlate with chemotherapy and the vaccine administration 
  5. Interpretation of the results - the complete response to treatment was after administration of chemotherapy together with the vaccine.  How can the authors tell that the response is from the vaccine and not from chemotherapy alone ? 

Author Response

This is an exciting paper demonstrating the use of a personalised dendritic cell vaccine used with peptide antigens in a patient with endometrial carcinoma.  The idea is novel and represents a promising new treatment paradigm for metastatic endometrial cancer.

Specific comments

  1. It appeared that the vaccine did not prolong time to recurrence as the patient recurred soon after primary treatment despite the administration of the vaccine.  However, a complete response was seen when used with chemotherapy.  Would the authors attempt to explain this and how would this correlate with the immune response generated ?

We thank the reviewer for their question. The patient recurred 11 months after her primary treatment. This is a respectable PFS base on the highly aggressive endometrial carcinoma setting. Her vaccination started long time after the primary chemotherapy and radiotherapy combination treatment due to vaccine production.

Based on the available data form this single patient, we provided the evidence the vaccine elicited an anti-tumor immune response that probably enhanced the anti-tumor effect of the second line chemotherapy association. The observation of the vaccination derived immune response form the rest of the study data set would allow a more scientifically solid explanation.

  1. In the results presentation, it would be better if the authors could describe in more details on the sites of recurrence and the size of the recurrent lesions. The PET-CT images in Fig 2A  showed a snap shot of the sites of recurrence but it would be better if these could be described in the text.  In addition, the background colour for Fig 2A and 2B were different,  and this would make the results less convincing.  The anatomical structures that the arrows were pointing to also needed to be double checked. 

We thank the reviewer for this comment. The following information has been added regarding the sites of recurrence: relapse localized to the previous irradiated left internal iliac lymph nodes and the meso-rectal lymph nodes.

The PET images in figure 2A and B were acquired on two different PET-CT machines with different software/hardware. The FDG intake is concordant between the two PET-CT machines. We updated the arrows in Fig 2A and 2B as they had been displaced.

  1. More description of the toxicities experienced by the patient would be useful.  

We thank the reviewer for this comment. We have added the following sentence to section 3.1 to clarify the toxicities: The observed hematological grade 3 toxicities concerned neutropenia occurring on day 8 and day 7 of the fourth and fifth chemotherapy cycles.

  1. For Fig 2C, it would be helpful to indicate when the chemotherapy was given as well, so that the reader could have some better idea of how CA 125 response correlate with chemotherapy and the vaccine administration 

We agree with the reviewer and have updated figure 2C to include the chemotherapy treatment window.

  1. Interpretation of the results - the complete response to treatment was after administration of chemotherapy together with the vaccine.  How can the authors tell that the response is from the vaccine and not from chemotherapy alone ? 

We agree with the reviewer that the observed oncological complete response can be the result of the chemotherapy and that the vaccine plus value to the effect is much more difficult to be confirmed base on a single patient data. We acknowledge this is a limitation of this paper and more patient’s data are needed in order to confirm this observation.

Nevertheless, in the recurrent SEC metastatic setting a complete response after chemotherapy is an infrequent outcome of chemotherapy-only therapeutic strategies. Furthermore, responses, when they occur, are often short-lived while in our patient, despite early interruption of chemotherapy, the PFS2 is longer than the PFS 1 (remission inversion) and the patient interestingly remains up to this date in complete response. This PFS remission-inversion is suggestive of a potential added anti-tumor effect from the concomitant vaccine that elicited a specific anti-tumor effect over the long term. More observation is needed to confirm this hypothesis.

Round 2

Reviewer 1 Report

The authors have addressed all comments